# Assessment of the Extent of Intracerebral Hemorrhage Using 3D Modeling Technology

**DOI:** 10.3390/healthcare11172441

**Published:** 2023-08-31

**Authors:** Joanna Chwał, Paweł Kostka, Ewaryst Tkacz

**Affiliations:** 1Department of Biosensors and Processing of Biomedical Signals, Faculty of Biomedical Engineering, Silesian University of Technology, 44-100 Gliwice, Poland; pawel.kostka@polsl.pl (P.K.); ewaryst.tkacz@polsl.pl (E.T.); 2Joint Doctoral School, Silesian University of Technology, 44-100 Gliwice, Poland

**Keywords:** 3D modeling, brain modeling, medical data analytics, ICH, neurology, segmentation

## Abstract

The second most common cause of stroke, accounting for 10% of hospital admissions, is intracerebral hemorrhage (ICH), and risk factors include diabetes, smoking, and hypertension. People with intracerebral bleeding experience symptoms that are related to the functions that are managed by the affected part of the brain. Having obtained 15 computed tomography (CT) scans from five patients with ICH, we decided to use three-dimensional (3D) modeling technology to estimate the bleeding volume. CT was performed on admission to hospital, and after one week and two weeks of treatment. We segmented the brain, ventricles, and hemorrhage using semi-automatic algorithms in Slicer 3D, then improved the obtained models in Blender. Moreover, the accuracy of the models was checked by comparing corresponding CT scans with 3D brain model cross-sections. The goal of the research was to examine the possibility of using 3D modeling technology to visualize intracerebral hemorrhage and assess its treatment.

## 1. Introduction

Due to ongoing technical advancements, sectors like three-dimensional (3D) modeling increasingly represent reality, enabling a wide range of applications. Building precise models of particular anatomical structures and pathologies, such as the brain or intracerebral hemorrhage, aids numerous medical operations, simplifies the work of doctors, and increases patient safety. Due to the increasing number of stroke cases, 3D modeling may prove to be a useful tool for detailed change visualization and treatment evaluation. According to the 2010 Global Burden of Disease study, the absolute number of hemorrhagic strokes worldwide (including ICH hemorrhage and subarachnoid hemorrhage) increased by 47% between 1990 and 2010 [1].

### 1.1. Intracerebral Hemorrhage

Intracerebral hemorrhage (ICH) is a devastating disease that refers to any bleeding within the intracranial vault, including the brain parenchyma and surrounding meningeal spaces. ICH is also the second most common cause of stroke, accounting for 10% of hospital admissions [2]. Risk factors for ICH include hypertension, cigarette smoking, diabetes mellitus, excessive alcohol consumption, male gender, older age and Asian ethnicity [3]. People with intracerebral bleeding have symptoms that correspond to the functions controlled by the area of the brain that is damaged by the bleed.

ICH is one of the main causes of stroke-associated death and disabilities, even though it only accounts for 10% of all hospital admissions related to stroke. At 30 days, the overall mortality rate exceeds 50%, and over half of all ICH-related deaths take place in the first 24 h following the initial bleed [4,5,6]. Less than 20% of survivors’ functional outcomes at six months are independent [7]. Intraventricular hemorrhage (IVH), which occurs in up to 40% of cases, is characterized by bleeding into the ventricles and is associated with obstructive hydrocephalus and a worsening prognosis [8]. A large hematoma volume (>30 mL), posterior fossa position, older age, and admission mean arterial blood pressure (MAP) >130 mm Hg are additional variables linked to poor prognosis [5,6].

Non-contrast computed tomography continues to be the golden standard of imaging techniques for the initial diagnosis of ICH, because it is fast and accessible [9,10]. Subarachnoid hemorrhage, ischemic stroke, and ICH are among the several intracranial pathologies that can be distinguished with a CT scan. In terms of size, surrounding edema, mass effect, intraventricular clot extension, and increased intracranial pressure, and can also demonstrate the degree of bleeding. Therefore, it can estimate the extent of the problem and thus determine if patients can be treated or need surgery. ICH can be recognized on CT scans because blood appears brighter than other tissues. However, due to the images being cross-sectional, it is harder for the medics to visualize the whole hemorrhage without 3D (three-dimensional) modeling technology [9,11].

### 1.2. Three-Dimensional Modeling

An object’s three-dimensional (3D) modeling can be considered as a comprehensive process that begins with data collection and concludes with an interactive 3D virtual model on a computer. A computer-generated 3D model exists only in the computer’s memory; it consists of virtual surfaces defined by a series of points on the Cartesian plane [12]. When describing the process of reconstructing an item, the term “3D modeling” is frequently used to refer to the simple process of turning a measured point cloud into a triangulated network (or “mesh”) or textured surface. In the graphic, vision, and photogrammetric fields, three-dimensional modeling of objects and scenes is a significant and ongoing research challenge. For a variety of purposes, including inspection, navigation, object identification, visualization, and animation, three-dimensional digital models are necessary [12].

Modeling effects are visible in movies, animations, video games, and even in medicine. For example, the segmentation of hematomas is crucial to clinical workflow, and 3D modeling enables objective measuring of the hematoma volume expansion over numerous scans and makes it possible to determine if a hemorrhage is still actively bleeding. In addition, it enables the assessment of the effectiveness of therapy and the improvement of prognosis, in combination with an appropriately selected treatment [13,14], which is crucial in this research.

### 1.3. Animation

Animation can be defined as creating the illusion of motion in inanimate images or objects. This is achieved by recording changes over time, which must be convincing for the viewer. Most animation of a 3D model is performed by setting an animation keyframe, which requires the creator to define the start and end points of an action or movement. The computer then calculates the frames needed to visualize the action between keyframes [15].

Animation of 3D models makes it possible to produce instructional videos on how to proceed during medical operations and how to interpret 3D anatomy, including bones and muscles, from 2D images [16], as well as educational materials, which, for example, can help patients comprehend their medical conditions, the medications they would need to take, or the treatments they might undergo better. This way, patients may feel better prepared for their appointment.

## 2. Methodology

The project was developed in the following free open source programs:3D Slicer (version 4.9.0)—obtaining 3D brain models from patients’ CT scans;Blender (version 2.78c)—model improvement and animation;Visual Studio 2017, using Windows Presentation Foundation (WPF), C# programming language and XAML—development of simply application;Autodesk Meshmixer—obtaining cross-sectional images;Jupyter, using the Python 3.6 programming language—model quality analysis.

The main stages of the project are shown in Figure 1.

### 2.1. Examination of Dataset and Possible Solutions

The data set used in this project consists of 15 medical examinations, obtained using CT, which belong to 5 anonymous patients (3 for each patient). The first examination refers to the diagnosis stage, the second was carried out after one week of treatment, and the third after two weeks. The data set was provided by the Department of Neurology in Zabrze, Medical University of Silesia. All images were formatted into a DICOM extension. The intracerebral bleeding can be seen in Figure 2.

Based on the available data set, three options were taken into consideration for the purpose of segmenting particular brain structures: manual segmentation with a semi-automatic region-growing algorithm, semi-automated segmentation using the *level tracing* tool and region-growing algorithm of the Slicer 3D program, and fully automated segmentation (using a deep-learning-based segmentation model such as UNet).

Manual CT scan segmentation refers to the process of manually tracing and annotating the different structures or tissues within a CT scan image. This is typically performed by a trained radiologist or medical image analyst using software tools specifically designed for this task. The goal of manual CT scan segmentation is to create a visual representation of the different structures within the image, allowing for further analysis and interpretation.

The manual segmentation process involves opening the CT scan image in the software and tracing the boundaries of the structures of interest. This can be done by clicking along the boundaries with a mouse or stylus or by using drawing tools within the software. The resulting segmented image can then be used for tasks such as disease diagnosis, treatment planning, and measurement of anatomical structures.

Semi-automated segmentation refers to a process in which the segmentation of an image or a volume is performed with a combination of automated algorithms and human interaction. The automated algorithms provide an initial segmentation that is then refined or corrected by a human operator. This approach allows for the benefits of both automated and manual segmentation, with the automated algorithms providing a quick and efficient initial segmentation, while the human operator can correct any errors or make adjustments to fit the requirements of a specific task better.

UNet is a deep-learning-based image segmentation model that is widely used for medical image analysis, including CT scans. UNet is an encoder–decoder architecture—the encoder part of the network is used to extract features from the input image and reduce its spatial resolution, while the decoder part of the network is used to restore the spatial resolution and produce the segmentation mask. The UNet architecture also uses skip connections between the encoder and decoder parts to improve the flow of information and to help the network learn fine details of the image.

To train the UNet model, annotated CT scan images are used as input, and the network is optimized to produce segmentation masks that closely match the ground truth annotations. Once trained, the UNet model can then be applied to new, unseen CT scans to produce segmentation masks for these images.

### 2.2. Determining the Project Assumptions

There are various significant factors that can affect the quality of the product: the time needed to create the model (whether the chosen method is time-consuming), interpretability of the results (whether the user can recognize structures and assess whether they are correct), the amount of data needed to obtain a solution, resources computing (hardware requirements), correctness of the mapping (accuracy), and complexity of the solution. These are established on the basis of the purpose of the research and available solutions.

### 2.3. Multi-Criteria Analysis

The simultaneous effect of all the provided assumptions on the ideal model was evaluated using a research method known as multi-criteria decision analysis (MCDA) to find the optimal solution. The given research applies the subjective weighted method, where each criterion’s weight denotes the importance of the criterion, depending on the nature of the problem and level of complexity. The simplest strategy, applied in many studies, is to give the criteria equal weights. However, the evaluation’s ultimate conclusions are significantly influenced by the criteria weights. The weights of the criteria are determined by the decision-makers’ preferences in subjective approaches [17,18,19]. The method is expressed as Formula (1) [20]:(1)$$F\left(x\right)=\sum_{i=1}^{k}w_{i}f_{i}\left(x\right),$$
where:*k*—number of objective functions;*x*—solutions vector;$w_{i}$—weighted coefficients; *i* = 1, …, *n*; expressed as Formula (2):(2)$$w\;\epsilon \;\left[0,1\right]\;and\;\sum_{i=1}^{k}w_{i}=1$$

For the purpose of obtaining an accurate assessment, each criterion was assigned a code (K1–K6). Evaluation of the criteria K1–K6 was the first step in the analysis, according to the relationship: 0—irrelevant, 0.5—equal, and 1—relevant (Table 1).

After summing up all the points (Table 1), the most relevant criterion was obtained, which is the accuracy.

The solution was evaluated (Table 2) on the basis of Table 1 in the following order:Column a—assessment of fulfillment of the criterion by each variant, assigning a value between 1 and 5 (where 1 does not meet the criterion);Column b—calculation of the significance ratio of the criterion with the value in column a;Variant evaluation—summed up values from column b;Value—the percentage value of the significance of the solution relative to the ideal solution.

On the basis of a multi-criteria analysis, solution number 1 was selected, which scored the most points in the criteria assessment and met the expectations in 78.125 % (Table 2). Concepts 2 and 3 received fewer points and were rejected due to:Solution 2—the worst accuracy, because of the semi-automatic segmentation. Using the *level tracing* tool, by moving the mouse, a user defines an outline where the pixels all have the same background value as the current background pixel, which, in the case of noise, results in the appearance of a significant number of misclassified elements and extends the time needed to create the model—various corrections must be made;Solution 3—the need to use a large number of images (the number of which in this project is limited)—the number of CT scans needed to train a UNet model for segmentation depends on several factors, such as the complexity of the structures to be segmented, the quality of the annotations, and the size of the network. In general, more data is better, as this allows the network to learn more comprehensive features. As a rough estimate, it is not uncommon to use hundreds or even thousands of CT scans to train a UNet model for medical image segmentation. Moreover, more computing power is needed than in the case of manual segmentation.

Image defects such as noise or artifacts can affect the accuracy and reliability of manual segmentation, but there are approaches that can reduce their impact in this process:Image processing—techniques such as checking the resolution of data set images, defining the region of interest (ROI), denoising, or filtering can be used to reduce noise or artifacts in the images before segmentation;Correction of topological errors—can be used to fill in missing data or resolve inconsistencies;Quality control—measures such as review of the segmentation results by an expert or similarity metrics can be used to identify and correct errors or inconsistencies in the segmentation process;Expert knowledge—expert knowledge about the anatomy or pathology being segmented can be used to inform the manual segmentation process and correct for any image defects or anomalies.

### 2.4. Designing 3D Models

Designing a 3D model in 3D Slicer involves several steps, shown in Figure 3. It is important to consider carefully the specific requirements of the task and to choose the appropriate segmentation and modeling tools based on the characteristics of the data and the desired outcome.

#### 2.4.1. Image Pre-Processing

Before designing a 3D model, the image data must be pre-processed to correct any artifacts or distortions and ensure that the data are suitable for modeling. In this project, pre-processing involved checking the resolution of the data set images, which was carried out to ensure that the images were based on isotropic voxels—the voxel shape is the same in every dimension (the *isotropic spacing* function was used for this purpose)—and defining the region of interest (ROI): a frame was set on the sections, the size of which was adjusted in such way that the images still contained the areas needed to create a model of the brain. After positioning the frame and checking that all structures of interest were inside it, areas that were not needed for further work (for example bones) were cropped out.

#### 2.4.2. Segmentation

Segmentation, in relation to images, is the process of separating uniform areas on the basis of a defined criterion [21], which involves identifying and isolating the regions of interest within the image data. In this work, the first step of segmentation was the creation of empty segments referring to specific brain elements (brain, ICH, ventricles, other—for example, bones) to be visible on the generated three-dimensional model (Figure 4).

The next step was to mark the structures corresponding to individual segments on the cross-sections using the selection tool—*Paint* (Figure 5), thanks to which it is possible to define voxels and assign them to a specific area. After selecting the segments, the semi-automated image segmentation function *Grow from seeds* was used, which causes the regions to grow based on the labels assigned to each anatomical structure and voxel intensity (Figure 6). The process of the *Grow from seeds* segmentation method can be summarized as follows:Initialization—the user selects one or more seed points in the image, which serve as starting points for the segmentation process;Segmentation growth—the algorithm then grows the segmentation region from the seed points based on some criterion, such as intensity, color, or texture;Termination—the segmentation process terminates when the region has reached a specified size, or when it reaches a boundary defined by a stopping criterion, such as an intensity gradient or a boundary in the image.


1. Load the image to be segmented



2. Set up the seed points for the region of interest



3. For each seed point:



    a. Initialize a new region with the seed point as the starting point



    b. While the region is growing:



        i. Determine the neighboring voxels of the current region



        ii. Calculate the similarity score between the neighboring



        voxels and the current region



        iii. Add neighboring voxels with similarity score above



        a threshold to the current region



        iv. Update the current region and repeat step (b)



4. Repeat step 3 for all seed points



5. Combine all the segmented regions into labeled volume



6. Save the segmented volume for further analysis or visualization


The *Grow from seeds* function uses the computer vision algorithm *GrowCut* for segmentation [22]. It is semi-automated, meaning that it requires some manual input from the user, but it also uses automatic processing to generate the final result. *GrowCut* creates clusters from predefined “seeds” according to membership validity. Images can be segmented according to the similarity of neighboring pixels. The method is implemented by two algorithms: the region-growing segmentation algorithm and the cellular automaton. The algorithm starts from user-selected grains (pixels) and implements a cellular automaton technique to segment homogeneous pixels in an image. Homogeneous grains are grouped according to their intensity in relation to the original pixel. The algorithm iteratively tries to check all pixels against inclusion criteria and stops when all pixels in the region of interest have the same intensity value and the same label [23].

One advantage of the *GrowCut* method is that it can handle complex structures and shapes, as well as non-homogeneous textures, making it well-suited for medical imaging applications. It is also relatively fast compared with other segmentation methods, and it can be easily integrated into other image-processing pipelines.

#### 2.4.3. Model Creation and Visualization

After segmenting the data, a 3D model can be created from the obtained regions. For this purpose, vectorization is used, which consists in changing the image described in the form of raster graphics (the image consists of a rectangular grid of points—pixels) into a vector image described with simple geometric figures, for example, points, lines, curves, or polygons [24].

In this paper, vectorization was carried out using the *closed surface function*, which transforms flat contours, formed during segmentation, into a mesh with a closed surface. The mesh consists of user-selected segments which, after vectorization, create a flat surface. After centering the scene, the mesh takes the form of a three-dimensional solid. The generated model is then improved to eliminate incorrectly formed layers during segmentation.

#### 2.4.4. Model Correction and Export

The created models need further correction, such as correction of topological errors, filling of small holes, and smoothing of the surface, which is why they were smoothed using the *joint smoothing* method (Figure 7). This method ensures that the segmented structures do not overlay in the smoothing process and is based on the standard method of image filtration—convolution operation using a triangular filter.

The convolution operation is a mathematical operation that is commonly used in image processing. It involves multiplying each element in a filter (also called a kernel) with the corresponding elements in an image, and summing the results to produce a new image. The filter is usually a small, fixed-size matrix that is used to modify the image.

A triangular filter is a type of filter that uses a triangular-shaped kernel to perform the convolution operation. In a triangular filter, the values of the kernel decrease linearly from the center of the kernel to its edges. This filter is commonly used for image smoothing, as it can reduce noise and other unwanted artifacts while preserving the overall structure of the image—pixel values are determined based on the weighted average of neighboring pixel values [22].

After smoothing, the three-dimensional model of the brain was saved in .stl (Standard Triangle Language) format and exported for use in other applications and further analysis.

#### 2.4.5. Model Refinement and Animation

Models were further improved using 3D graphics designer software (Blender). The noise in their structure was removed—the misclassified parts in the segmentation process were rejected (Figure 8). For this purpose, the *remesh* modifier was used—it generates a new surface that has the same shape as the input image, but with a more regular topology. The modifier has a *threshold* function, which determines the level of detail that is preserved in the new mesh when the original mesh is remeshed.

The *threshold* works by specifying the maximum allowed deviation between the original mesh and the new mesh. Any part of the original mesh that deviates more than the specified threshold will be modified during the remeshing process to bring it closer to the new mesh. A low threshold value will preserve more detail in the original mesh, while a high threshold value will result in more simplification of the mesh. The optimal value depends on the specific requirements of the project and the desired outcome.

Animation was the final stage of model design. In Blender, creating an animation is based on frames. The program automatically modifies the position and properties of objects by calculating the differences between the last and newly added frames. Adding a frame is carried out through the *LocRotScale* function, which allows the user to lock the position, size, and rotation of the object at the time selected by the user. Location (Loc) refers to the position of the object in the 3D space, specified as three coordinates (X, Y, Z), rotation (Rot) refers to the orientation of the object in the 3D space, specified as three angles (pitch, yaw, roll), and scale (Scale) refers to the size of the object and defines the scaling along each axis.

### 2.5. Developing the Application

The next step of this research was to develop a simple application that allows users to visualize and manipulate created models. There are several purposes for using applications to visualize 3D structures, including:Scientific and medical visualization—applications are used to visualize and analyze medical and scientific data, such as CT scans, MRI scans, and microscopy images. This allows researchers and medical professionals to understand complex 3D structures better and to identify patterns and relationships that are not easily apparent in 2D images;Education and training—applications are used in educational and training settings to visualize 3D structures and systems, such as anatomy and biology, and to help learners understand complex concepts better.

The application consists of four windows:
The first window, with the animated brain model, opens after starting the application and allows the user to go to the functional part of the program (Figure 9);Patient window—the user can select the patient’s brain model to be visualized (Figure 10);Selection window—this allows the user to go to the next application modules (Figure 11);The fourth window is designed to visualize and to manipulate the brain model and its components in three dimensions (Figure 12).

The Helix Toolkit 3D, an open-source 3D graphics library that enables users to work with 3D objects using Windows Presentation Foundation (WPF) technology, was the primary library used to construct the application. In the instance of this work, the library enables users to visualize 3D objects and rotate them in three dimensions.

Overall, applications for visualizing 3D structures provide a powerful tool for exploring and understanding complex 3D data, enabling users to make more informed decisions and to communicate their findings more effectively.

### 2.6. Model Quality Analysis

An analysis of the similarity (Table 3) was performed to check the differences between the designed brain models and the anatomical brains (CT scans). Two popular similarity metrics were used to assess the quality of the models: the Sørensen–Dice coefficient and the Hausdorff distance, which are both used in studies focused on, for example, abdominal multi-organ auto-segmentation [25] or whole-body-organ segmentations [26]. Comparing images to assess segmentation accuracy is a key part of assessing progress in this area of research [27].

In order to provide a project with metrics to evaluate similarity, a cross-section from obtained brains was made and compared with original CT scans. By using the Jupyter environment and the Python programming language, which was used to create the script required for the computations, the similarity analysis was completed. An example of the adjustment is shown in Figure 13. In addition, the standard deviation for the obtained results was calculated.

The Sørensen–Dice coefficient, also called the overlap index, is the most commonly used index in the validation of volumetric segmentation, and indicates the extent to which the segmentation mask overlaps the original scans, defined by the formula [27]:(3)$$DICE=\frac{2|A\cap B|}{\left|A|+|B\right|}$$
where $\frac{2|A\cap B|}{\left|A|+|B\right|}$ is the doubled sum of elements common to both sets divided by the sum of the number of elements in each set.

Pseudocode implementation of the Sørensen–Dice coefficient in Python programming language:


def dice_coefficient(A, B):



   A = set(A)



   B = set(B)



   intersect = A.intersection(B)



   return 100 * 2 * len(intersect) / (len(A) + len(B))


In this implementation, **A** and **B** are the two sets of data to be compared, and **intersect** is the set of elements that are present in both **A** and **B**. The Sørensen–Dice coefficient is calculated as the ratio of twice the size of the intersection of **A** and **B** to the total size of both sets. The final result is multiplied by 100 to convert it to a percentage. The returned value ranges from 0 to 100, with higher values indicating greater similarity between the sets.

The Hausdorff distance specifies the similarity between two images, and measures the shift between the segmentation mask and original scans. The measure is defined as the maximum value of the two functions h (A, B) and h (B, A):(4)H(A,B)=max(h(A,B),h(B,A)),
where $h\left(A,B\right)=max_{a\epsilon A}\left(max_{b\epsilon B}d\left(a,b\right)\right)$ denotes the greatest distance, d, of set A to the closest element of set B [28].

Pseudocode implementation of the Hausdorff distance in Python programming language:


def hausdorff_distance(A, B):



    def distance(a, b):



        return max([abs(a[i]-b[j]) for i in range(len(a)) for j



        in range(len(b))])



    return max(distance(A,B), distance(B,A))


In this implementation, **A** and **B** are the two sets of points to be compared, and **distance(a, b)** is a function that calculates the maximum distance between points in **a** and points in **b**. The Hausdorff distance is defined as the maximum of the two distances, **distance(A,B)** and **distance(B,A)**. The resulting value will be a non-negative real number, with higher values indicating greater dissimilarity between the sets.

Standard deviation is a measure of the amount of variation or dispersion in a set of data. It is calculated as the square root of the variance, which is the average of the squared differences from the mean:(5)$$\sigma =\sqrt{\frac{{\left(x_{1}-\bar{x}\right)}^{2}+{\left(x_{2}-\bar{x}\right)}^{2}+...+{\left(x_{n}-\bar{x}\right)}^{2}}{n}}$$
where:*x*—individual value;$\bar{x}$—the arithmetic mean of the values;*n*—number of values.

In other words, standard deviation measures how much the data points in a distribution deviate from the mean value. If the standard deviation is small, it means that the data points are clustered closely around the mean, indicating low variability. If the standard deviation is large, it means that the data points are spread out over a wider range, indicating high variability.

The lower the values of the Hausdorff distance and standard deviation are, and the higher the value of the Sørensen–Dice similarity coefficient is, the better is the representation of the obtained models. Since the differences between the obtained values are negligible, it can be said that the model was accurately mapped and its quality analysis was successful.

The volume of intracerebral hemorrhage was calculated on the obtained 3D models after diagnosis, and after one week and two weeks of treatment, using the Slicer 3D Segment Statistic module, which takes into account pixel size, slice spacing and axis directions (Table 4). After a week, the volume of the ICH for most patients was reduced by half and after two weeks it was not detectable.

After 2 weeks, intracerebral haemorrhage was not detectable, and only brain edema was visible.

## 3. Discussion

The results presented in this study prove the high accuracy of mapping patients’ brains using 3D modeling methods, which enables the determination of the volume of ICH, the analysis of the treatment effectiveness, and the recovery time. Brain models enable the visualization of pathological areas in three dimensions, thanks to which they allow for more accurate analysis and faster selection of the appropriate treatment method, and the created application allows for quick access to selected anatomical structures from specific patients. The following were identified as the key modeling objectives in medical usage during the creation of this paper: collecting data on pathological states and extracting data on anatomical structures, lesions, and diseases to build a database of anatomical structures for clinical diagnosis, in order to choose a treatment strategy for therapy, distinctive parameters must be measured, as well as the location, shape, and distribution of the lesion structure, each section after segmentation is distinct, and users can pick amongst them to concentrate on those specific areas.

In addition to the segmentation method described in this article, there are also other methods in Slicer. One of them is the threshold method. CT scans respond well to threshold segmentation and the algorithm is simple and fast. However, the user must make a judgment based on experience or perform many trial segmentations before making a threshold adjustment [21]. The second one is the FCM algorithm, which is associated with a long computation time. Segmentation of smooth-edged areas takes less time to complete; however, if the edge smoothness is insufficient, the number of interactions required to complete accurate segmentation will increase and the segmentation process will take longer overall. This approach can produce better image segmentation results, retain much detailed information, and create an edge area that is very clear and more like the actual block area [21]. Both methods are faster and semi-automatic; however, they are not effective for overly noisy DICOM images because they incorrectly assign specific areas to the labels.

Three-dimensional visualization can produce great results in diagnosing diseases and assessing the extent of lesions; however, medical professionals can use 3D-printed models for a variety of purposes, including accurate replication of anatomy and pathology for pre-surgical planning and simulation of complicated surgical or interventional procedures, as well as to educate patients and medical students and to improve doctor–patient communication. Furthermore, developing the best CT scanning techniques can be done affordably by using patient-specific 3D-printed models. In order to understand the effects of 3D printing on clinical decision-making and patient outcomes better, current research should go beyond studies of model correctness [29].

In articles, researchers have also created models of other anatomical structures for a variety of purposes. These include a heart model to create a dynamic, three-dimensional model, that can be used for educational, visualization purposes, for the purpose of directing stent graft placement and assessing coronary lumen stenosis brought on by calcified plaques, replicating vascular diseases such as aortic/cerebral aneurysm, pre-surgical planning and simulations [29,30,31,32]; a 3D model of vertebrae, which will result in faster and more diverse spine research, because scientists and researchers rely on cadaver specimens to try new treatment methods, and because the number of people willing to donate their bodies for research purpose is very low or nonexistent in most countries [33]; diagnostic evaluation of tumors in terms of their size, extent, and location, which is crucial for surgical planning and making decisions about tumor resection [29]; and patient-specific 3D-printed models based on imaging data serve as a reliable tool to develop optimal CT protocols with low radiation doses and acceptable image quality [29,34].

Despite the fact that image-based modeling can create accurate and realistic-looking models that are sometimes even comparable to models produced by laser scanning, it still requires a large amount of human interaction because the majority of the current automated methods have not yet been tested in practical settings. Therefore, it is still a challenging problem and a popular research area to create geometrically accurate, full, and detailed 3D models of complex objects. Full automation is still out of reach if the objective is to produce accurate, comprehensive, and photo-realistic 3D models of medium- and large-scale items in realistic settings. It is challenging to choose automated procedures for practical applications because it is typically impossible to evaluate their effectiveness properly [12].

## 4. Conclusions and Outlook

The goal of research was to examine the possibility of using 3D modeling technology to map and visualize intracerebral hemorrhage. The models were created successfully and in the process of evaluating similarity they gave acceptable results.

There are several advantages of manual segmentation, including: flexibility—allows the operator to adapt to the specific requirements of a particular task and to make adjustments as needed; high accuracy—the operator has full control over the segmentation process, allowing them to ensure a high degree of accuracy; interpretable results—results can be easily interpreted and understood, making it easy to assess the quality of the segmentation and to diagnose and correct errors; reproducibility—results can be easily reproduced and compared with results from other studies, allowing for consistency and comparability across different data sets; and human expertise—can take advantage of the human operator’s knowledge, intuition, and experience, allowing for a level of accuracy and detail that is difficult to achieve with automated methods. However, all three considered solutions have their disadvantages:Manual segmentation: time-consuming—can be a slow and labor-intensive process, especially for large and complex models. Lack of reproducibility—can be subjective and may vary from operator to operator, making it difficult to reproduce results and compare results from different studies. Limited scalability—not well-suited for large-scale studies or for creating models for very large data sets, as it can become impractical to perform manually for such large data sets. Operator dependence—the quality of the manual segmentation can depend heavily on the skill and experience of the operator, making it difficult to ensure consistent results across different studies and operators;Semi-automated segmentation: dependence on initial segmentation—the quality of the final segmentation result depends heavily on the quality of the initial segmentation provided by the automated algorithm. If the initial segmentation is poor, it can be difficult to correct or improve it through human interaction. Limited automation—while semi-automated segmentation provides some benefits of automation, it still requires human interaction to refine the segmentation. This can still be time-consuming and may not always be practical for large data sets or for situations where speed is of the essence. User bias—the human operator can influence the final segmentation result, potentially introducing bias or subjective judgment into the process. This can affect the accuracy and reproducibility of the segmentation. Lack of standardization—there is currently no standard method for performing semi-automated segmentation, making it difficult to compare results from different studies and to ensure consistent results across different operators;Fully automated segmentation: lack of flexibility—automated algorithms may not be able to adapt to variations in the data or to the specific requirements of a particular task, leading to sub-optimal results in some cases. Limited accuracy—automated algorithms can be limited by the quality and availability of training data, and may not always produce accurate results. In some cases, manual correction may still be necessary to obtain an acceptable level of accuracy. Uninterpretable results—automated algorithms can be difficult to interpret and understand, making it challenging to assess the quality of the segmentation and to diagnose and correct errors. Lack of control—with fully-automated segmentation, the user has limited control over the segmentation process, making it difficult to make specific adjustments or to correct errors. Generalizability—automated algorithms may not generalize well to new data or to different imaging modalities, leading to sub-optimal results in these cases. Computational resources—automated segmentation algorithms can be computationally intensive, requiring significant processing power and memory, making it challenging to perform on large or complex data sets.

The scientific novelty of the study lies in the use of technology to provide a more accurate and comprehensive assessment of intracerebral hemorrhage. Traditionally, assessment of the extent of ICH has been carried out through imaging techniques such as computed tomography (CT) scans or magnetic resonance imaging (MRI). However, 3D modeling technology offers a new approach to this assessment by allowing for the creation of a detailed 3D model of the brain that can be used to visualize the extent and location of the hemorrhage in a more precise and interactive manner. Traditional two-dimensional (2D) and three-dimensional (3D) visualization tools, despite promising results in the literature, are still restricted to a 2D screen, which has an impact on the realistic visualization of anatomical structures and pathologies of 3D data sets. This is especially apparent when dealing with complex pathologies. In order to identify the regions of interest (desired anatomical structures and pathologies) from volumetric data, a sequence of image postprocessing and segmentation stages are typically used to create patient-specific 3D-printed models from a patient’s CT or MRI imaging data [29]. The models created in this study are completely 3D-printable and can be used to plan surgeries, treatments, or instruct aspiring and practicing doctors.

The assessment of intracerebral hemorrhage using a printed 3D model of a brain and other anatomical structures can be of great assistance to neurologists and neurosurgeons. It will be helpful in the accurate estimation of an anatomical structure affected by ICH, and assessing the extent of an area with hemorrhage and the results of treatment. It can provide medical personnel with the information about the quality of the treatment and volume of the changes in every stage of the bleeding.

In the future, the study might be expanded to examine the similarity of more models in order to determine with higher accuracy the possibility of creating an anatomical model of the brain and using graphical tools to represent the human brain accurately. In addition, machine learning may be used to provide doctors with automatic methods for segmenting ICH lesions from CT scans.

## Figures and Tables

**Figure 1 healthcare-11-02441-f001:**
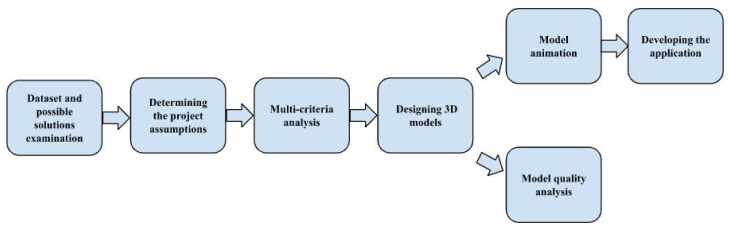
Main steps of the project.

**Figure 2 healthcare-11-02441-f002:**
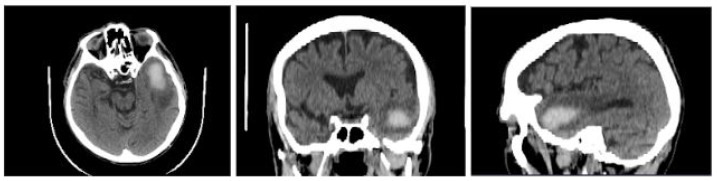
Images of a patient with ICH in axial (on the **left**), coronal (in the **middle**), and sagittal (on the **right**) planes.

**Figure 3 healthcare-11-02441-f003:**
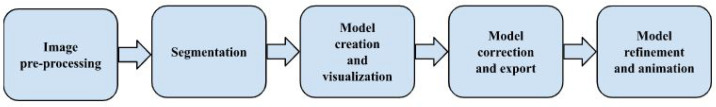
Main steps of the 3D models designing.

**Figure 4 healthcare-11-02441-f004:**
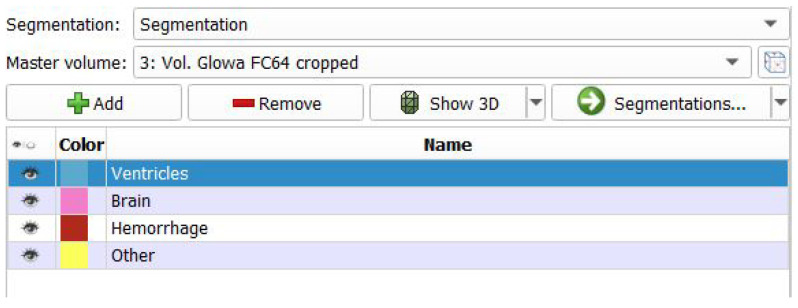
The creation of empty segments.

**Figure 5 healthcare-11-02441-f005:**
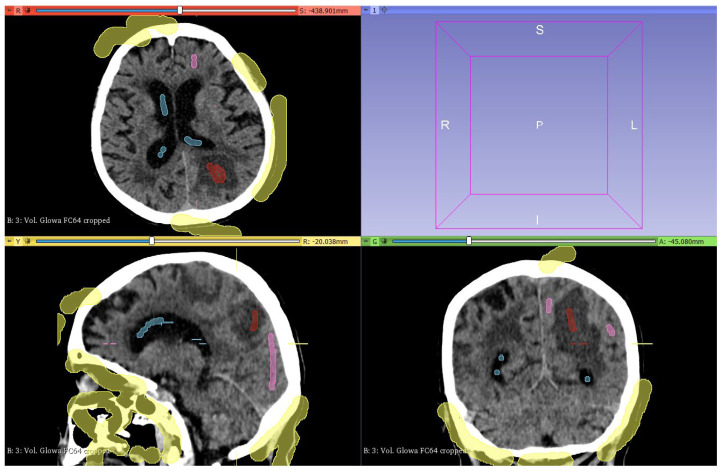
Marked brain structures (red color—ICH, pink—brain tissue, blue—ventricles, and yellow—others).

**Figure 6 healthcare-11-02441-f006:**
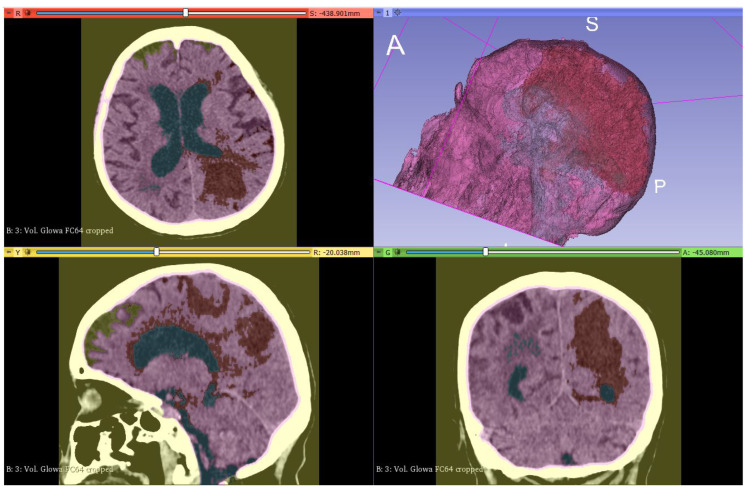
Segmentation process after using *Grow from seeds* function.

**Figure 7 healthcare-11-02441-f007:**
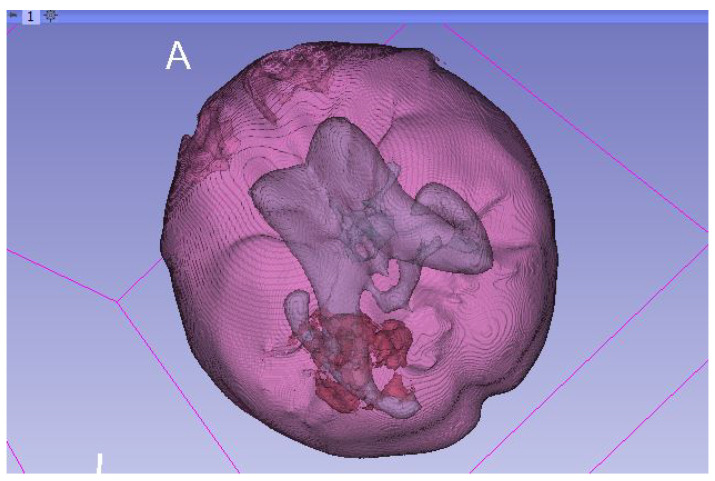
Visualization of the created model after applying the *joint smoothing* method.

**Figure 8 healthcare-11-02441-f008:**
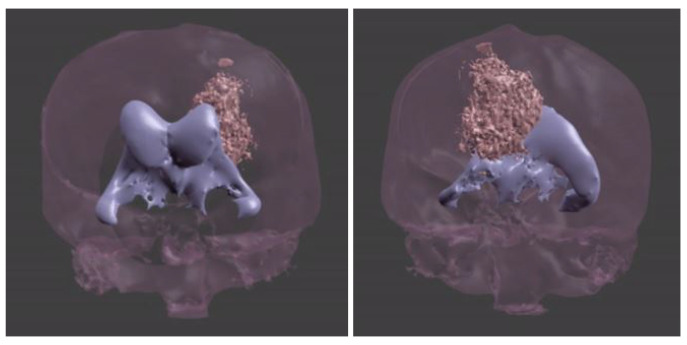
3D model after remeshing process.

**Figure 9 healthcare-11-02441-f009:**
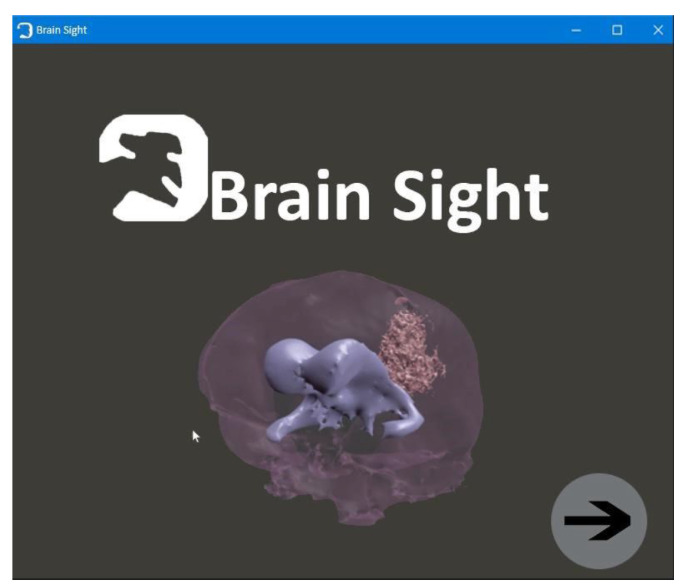
First window.

**Figure 10 healthcare-11-02441-f010:**
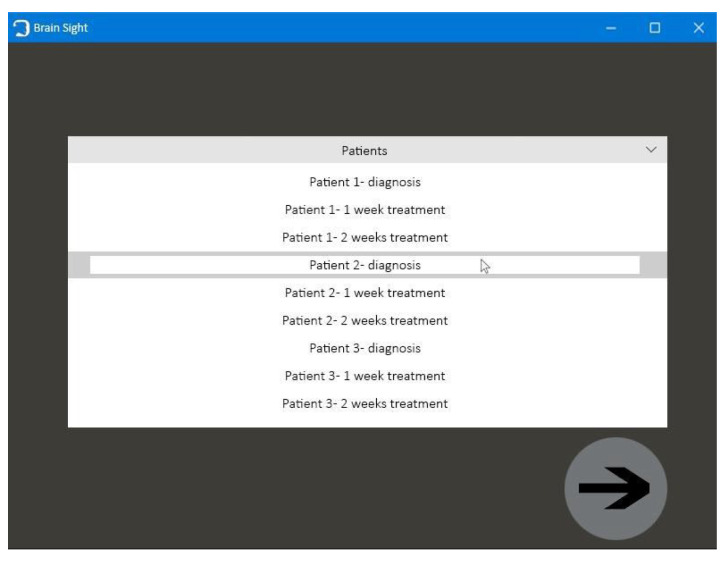
Patient window.

**Figure 11 healthcare-11-02441-f011:**
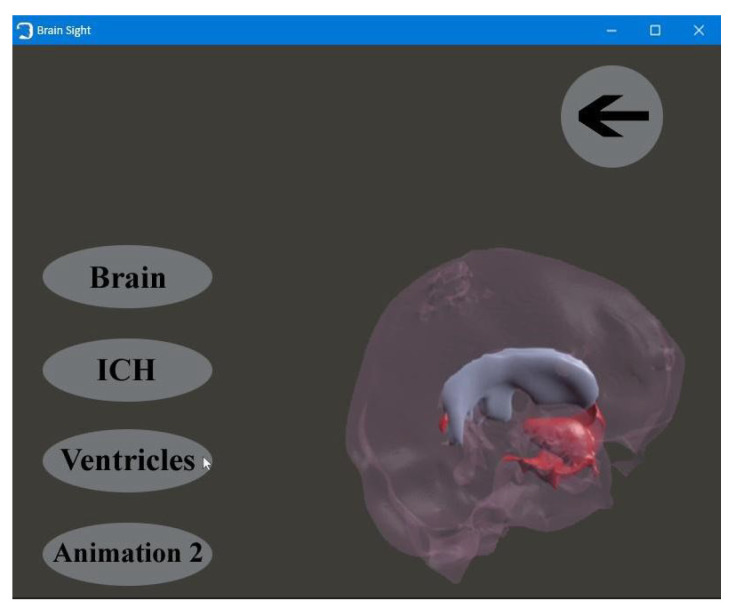
Selection window.

**Figure 12 healthcare-11-02441-f012:**
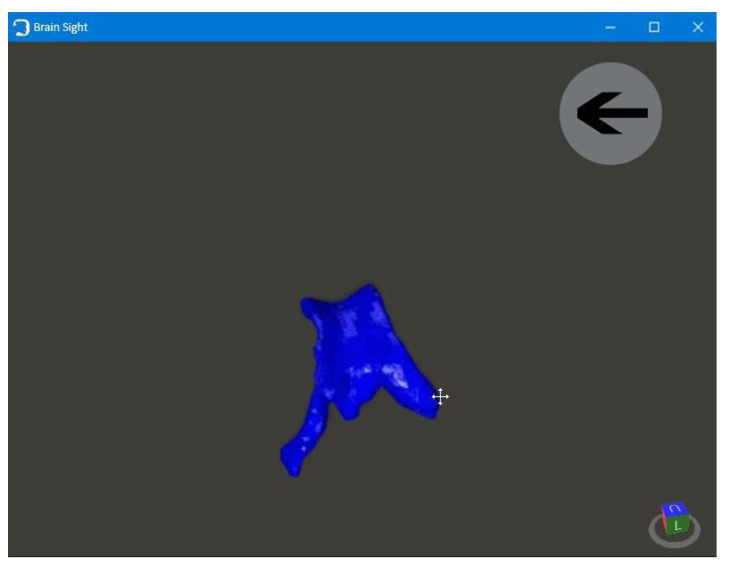
Fourth window.

**Figure 13 healthcare-11-02441-f013:**
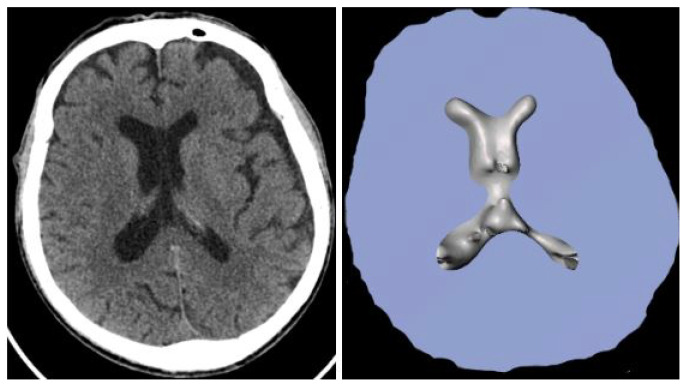
Original CT scan (on the left side) and cross-section of obtained brain model (on the right side).

**Table 1 healthcare-11-02441-t001:** Materiality criteria, where: K1—time-consuming, K2—interpretable results, K3—the amount of data needed to obtain a solution, K4—computational resources, K5—accuracy, and K6—complexity.

	K1	K2	K3	K4	K5	K6	SUM
K1	X	0.5	0.5	1	0	0.5	2.5
K2	0.5	X	1	1	0	1	3.5
K3	0.5	0	X	0.5	0	0.5	1.5
K4	0	0	0.5	X	0	0.5	1
K5	1	1	1	1	X	1	5
K6	0.5	0	0.5	0.5	0	X	1.5

**Table 2 healthcare-11-02441-t002:** Evaluation of solutions (solution 1—manual segmentation with a semi-automatic region-growing algorithm, solution 2—semi-automatic segmentation using the *level tracing* tool and region-growing algorithm of the Slicer 3D program, solution 3—the use of a deep-learning-based segmentation model, for example UNet).

Criteria	Materiality Criteria	Ideal Solution		Solution 1		Solution 2		Solution 3	
		**a**	**b**	**a**	**b**	**a**	**b**	**a**	**b**
**K1**	**2.5**	5	12.5	3	7.5	2	5	4	10
**K2**	**3.5**	5	17.5	5	17.5	5	17.5	2	9
**K3**	**1.5**	5	7.5	5	7.5	5	7.5	1	1.5
**K4**	**1**	5	10	4	4	4	4	3	3
**K5**	**5**	5	25	4	20	1	5	3	15
**K6**	**1.5**	5	7.5	4	6	4	6	2	3
**Variant evaluation**		80		62.5		45		41.5	
**Value**		100%		78.125%		56.25%		51.875%	

**Table 3 healthcare-11-02441-t003:** Sørensen–Dice similarity coefficient and Hausdorff distance results.

	Sørensen–Dice Similarity Coefficient [%]	Sørensen–Dice Similarity Coefficient—Standard Deviation	Hausdorff Distance [mm]	Hausdorff Distance—Standard Deviation
Model 1	91		3.20	
Model 2	92		3.23	
Model 3	94	1.17	2.33	0.44
Model 4	94		2.15	
Model 5	93		2.60	

**Table 4 healthcare-11-02441-t004:** Volume of intracerebral hemorrhage.

	ICH Volume [cm${}^{3}$] after First Diagnosis	ICH Volume [cm${}^{3}$] after a Week	Approximate Percentage Change of ICH Volume after Week [%]
Patient 1	17.0607	7.7141	−55
Patient 2	76.4668	33.1258	−57
Patient 3	17.5321	8.2503	−53
Patient 4	8.5965	0	−100
Patient 5	52.9358	26.0897	−51

## Data Availability

The data presented in this study are available on request from the corresponding author.

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
