# Peer review of "Assessment of the Extent of Intracerebral Hemorrhage Using 3D Modeling Technology"

_healthcare, 2023, doi:10.3390/healthcare11172441_

Round 1
Reviewer 1 Report (Previous Reviewer 1)
The article is devoted to solving the problem of visualization in medical diagnostics. The topic of the article is relevant. The level of English is acceptable. The article is easy to read. The figures in the article are of acceptable quality. The article cites 36 sources, some of which are not relevant. The References section is sloppy.
The following comments and recommendations can be formulated on the material of the article:
1. At the very beginning of 2020, Google’s artificial intelligence department called DeepMind introduced a deep learning model that outperformed the average radiologist by 11.5% and significantly reduced the workload of a second specialist in the British research system. Another recent study conducted by university hospitals in South Korea showed that AI has a higher sensitivity in detecting cancer compared to living specialists, especially in the case of breasts with a large fat layer (90% vs. 78%). Research is still in its early stages and needs more clinical trials. At the moment, models can serve as an additional specialist, automatically issuing a second opinion. Potentially, they can fill the growing shortage of qualified radiologists. By the way, how did the authors substantiate the issue of clinical trials in their research? Do the authors have the appropriate licenses and rights?
2. Breast Health Solutions by iCAD (New Hampshire, USA; FDA approved and EU compliant). The AI package applies deep learning algorithms to 2D mammography, 3D mammography (digital breast tomosynthesis, DBT), and breast density assessment. Its ProFound AI technology was the first FDA-approved AI solution for 3D mammography. Transpara by ScreenPoint Medical (Netherlands; FDA-approved and EU-compliant solution). Trained on over a million mammograms, Transpara's deep learning algorithm helps radiologists analyze 2D and 3D mammograms. The solution is already in use in 15 countries, including the US, France and Turkey. And who approved or supported the author's research?
3. Finally, in March 2020, the Journal of Investigative Dermatology published a study by scientists from Seoul National University. Their CNN model was trained on more than 220,000 images to predict malignancy and classify 134 skin diseases. AI also confirmed its ability to distinguish between melanoma and birthmarks at the level of a living specialist. In addition to improving the speed and accuracy of diagnostics, there are plans to implement CNN algorithms on smartphones for non-professional skin examination. This can push people to visit dermatologists for damage they would otherwise ignore. And what devices is the author's proposal focused on? What is the reason for this choice?
4. Despite all the promising research, none of the cancer detection programs are currently FDA-approved for the North American market due to potential harm from poor quality diagnostics. At the same time, two melanoma detection solutions have received EU approval, meaning they meet EU safety standards. Well, you understand, dear authors?
-
Author Response
Dear Sir/Madam,
Thank you very much for your review.
1 & 2
We are aware of the progress and capabilities of artificial intelligence. The research, detailed in the review, is geared towards finding lesions, which of course is important for diagnosis and treatment, but in doing so it does not segment entire organs, which in some cases can be extremely important. I am talking here, for example, about the issue of later printing a given model, in order to practice before surgery. I have personally collaborated, among other things, in creating a model of the heart before cardiac surgery on a newborn. The model was used by doctors to plan the operation in detail and consider possible options.
Based on research, we know that the computer is better at recognizing pathologies, but we do not have enough data to assess how it behaves in terms of creating a model of a single organ with the lesion and preparing it for 3D printing.
Unfortunately, we do not have a license, while the work was consulted with the data provider - Department of Neurology in Zabrze, Medical University of Silesia.
In the past I have been working on developing an algorithm for early detection of multiple sclerosis lesions. Of course, a computer can be more accurate than a radiologist, and this is due in part to the issue of distinguishing much more grayscale colors. However, the algorithm will not always be prepared for anomalies, comorbidities, or more complex cases.
3. The models created using our method can be used both in education, such as in student classes, and in the preparation of a treatment plan or surgery by doctors. The reasons for the choice are efficiency, the capabilities of the chosen solution, as well as cost, since it does not require the purchase of expensive software to work on a particular case, it is based on open source.
4. Of course, we understand and do not dispute the arguments given. Our solution is based more on single orders in creating comprehensive models of the organs and pathologies in question, rather than assessing the presence of lesions alone.
Kind regards,
Joanna Chwał
Reviewer 2 Report (Previous Reviewer 2)
Authors have adjusted most of the changes.
Therefore, this manuscript can be accepted in its present form.
Author Response
Dear Sir/Madam,
Thank you very much for your review.
Kind regards,
Joanna Chwał
Round 2
Reviewer 1 Report (Previous Reviewer 1)
I formulated the following comments to the basic version of the article:
1. At the very beginning of 2020, Google’s artificial intelligence department called DeepMind introduced a deep learning model that outperformed the average radiologist by 11.5% and significantly reduced the workload of a second specialist in the British research system. Another recent study conducted by university hospitals in South Korea showed that AI has a higher sensitivity in detecting cancer compared to living specialists, especially in the case of breasts with a large fat layer (90% vs. 78%). Research is still in its early stages and needs more clinical trials. At the moment, models can serve as an additional specialist, automatically issuing a second opinion. Potentially, they can fill the growing shortage of qualified radiologists. By the way, how did the authors substantiate the issue of clinical trials in their research? Do the authors have the appropriate licenses and rights?
2. Breast Health Solutions by iCAD (New Hampshire, USA; FDA approved and EU compliant). The AI package applies deep learning algorithms to 2D mammography, 3D mammography (digital breast tomosynthesis, DBT), and breast density assessment. Its ProFound AI technology was the first FDA-approved AI solution for 3D mammography. Transpara by ScreenPoint Medical (Netherlands; FDA-approved and EU-compliant solution). Trained on over a million mammograms, Transpara's deep learning algorithm helps radiologists analyze 2D and 3D mammograms. The solution is already in use in 15 countries, including the US, France and Turkey. And who approved or supported the author's research?
3. Finally, in March 2020, the Journal of Investigative Dermatology published a study by scientists from Seoul National University. Their CNN model was trained on more than 220,000 images to predict malignancy and classify 134 skin diseases. AI also confirmed its ability to distinguish between melanoma and birthmarks at the level of a living specialist. In addition to improving the speed and accuracy of diagnostics, there are plans to implement CNN algorithms on smartphones for non-professional skin examination. This can push people to visit dermatologists for damage they would otherwise ignore. And what devices is the author's proposal focused on? What is the reason for this choice?
4. Despite all the promising research, none of the cancer detection programs are currently FDA-approved for the North American market due to potential harm from poor quality diagnostics. At the same time, two melanoma detection solutions have received EU approval, meaning they meet EU safety standards. Well, you understand, dear authors?
My comments were aimed at ensuring that the authors, answering them, would reveal the applied value of their research. I saw that the authors sufficiently understood my idea and implemented it in the current version of the article. I support. Publication of this article. I wish the authors creative success.
-
This manuscript is a resubmission of an earlier submission. The following is a list of the peer review reports and author responses from that submission.
Round 1
Reviewer 1 Report
As the authors state: “The goal of research was to examine the possibility of using 3D modeling technology to visualize intracerebral hemorrhage. The models were created successfully and in the process of evaluating similarity they gave acceptable results.” Thus, the article is devoted to the study of the possibility of using 3D modeling for the visualization of medical images. This topic is as old as a mammoth and as relevant as a mammoth. But everyone wants to see a live mammoth ) The structure of the article is classical. The level of English is acceptable. The figures in the article are of unsatisfactory quality. Quality Fig. 4, 5, 9-11 is completely unacceptable. The article cites 33 sources, most of which are not relevant.
The following remarks can be made on the material of the article:
1. The authors position their article as a research one, but they do not even bother to formulate the scientific novelty, object and subject of research. The design of the study would also be very appropriate. In addition, the topic is very hackneyed. A thoughtful, multi-criteria analysis of analogues with a list of their shortcomings is categorically necessary.
2. To be honest, it is not clear from the article what kind of result the authors got. I will proceed from the assumption that they were solving an image segmentation problem (what the authors say about 3D just makes the segmentation problem voluminous). So my hypothesis is built on the basis of formula (1) (this is one of the two formulas presented in the technical article). The most popular approach for the segmentation problem is to take some kind of neuroarchitecture (for example, Unet) with a pre-trained encoder and minimize the sum of $BCE$ (binary cross-entropy) and $DICE$ (Sørensen–Dice coefficient). The authors work with medical images. They are characterized by the presence of many defects. What problems do they create?
2a. Even if it seems to us that there are a lot of defects in a photograph, that it is very “dirty”, the area occupied by defects is still much smaller than the undamaged part of the image. To solve this problem, you can increase the weight of the positive class in $BCE$, and the optimal weight will be the ratio of the number of all clean pixels to the number of pixels belonging to defects.
2b. The second problem is that if we use Unet out of the box with a pre-trained encoder, like Albunet-18, we lose a lot of positional information. The first layer of Albunet-18 consists of a convolution with kernel 5 and stride equal to two. This allows the network to work quickly. We sacrificed network time for better defect localization: we removed max pooling after the first layer, reduced stride to 1, and reduced the convolution kernel to 3.
2c. If we work with small images, for example, compressing the picture to 256 x 256 or 512 x 512, then small defects will simply disappear due to interpolation. Therefore, you need to work with a large picture.
2d. During training, many pictures are placed on one card. Because of this, the estimation of the mean and variance in BatchNorm layers is inaccurate. In-place BatchNorm helps us solve this problem, which, firstly, saves memory, and secondly, it has a Synchronized BatchNorm version that synchronizes statistics between all cards.
3. However, judging by "3D Slicer (version 4.9.0), Blender (version 2.78c), Visual Studio 2017, Autodesk Meshmixer, Jupyter", we are not talking about intelligent image processing. The authors take a lot of images and build a 3D model "as God wills." I can't say otherwise, because the Methods section resembles a brochure, where the authors list the names without understanding the essence and rationality of building the model itself.
Author Response
Dear Sir/Madam
Thank you very much for your review, based on it we have made detailed revisions to the article:
- We added multi-criteria analysis, advantages/disadvantages of the solutions and changed the structure of the article.
- and 3. The results and "methods" section has been corrected.
Due to the fact that a large part of the article has changed, I am sending the entire article as an attachment.
Kind regards,
Joanna Chwał

Reviewer 2 Report
My few concerns are below.
1. What is the novelty of this manuscript?
2. I dont see any contributioms of this manuscript.
3. There is no comparison in this manuscript.
4. Authors must explain how they address the research gap while conductiong this research.
5. Authors should include a pseudo code of their developed and used method.
6. Manuscript lacks mathematical analysis of proposed method.
Author Response
Dear Sir/Madam
Thank you very much for your review, based on it we have made detailed revisions to the article:
1. Possibility to distribute 3D printed models of the brain (or other structures) to plan surgeries, treatments or to instruct aspiring and practicing doctors.
2. and 4. We've made major revisions throughout the article.
3. and 5. It has been added.
Due to the fact that a large part of the article has changed, I am sending the entire article as an attachment.
Kind regards,
Joanna Chwał

Reviewer 3 Report
Authors proposed a novel pipeline to evaluate the treatment of intracerebral hemorrhage. I have some doubts and hope authors can address:
1.There are many grammar errors and typos. Also the manuscript is not organized very well. Please revise the manuscript. For example, Line 111 should be Figure 4, not 4. The name of Figure 9 should be window not widow.
2.Legend of figures should include more details to explain. For example, what are these subfigures in Figure 1?
3.Can authors explain how the original CT scans are performed. Why this can be treated as the label for evaulation?
4.What is the semi-automatic algorithm mentioned in abstract? It seems not mentioned in the manuscript.
5.Authors have used 5 patients to evaluate. Is this small data enough to show the robustness for the the proposed approach?
Author Response
Dear Sir/Madam
Thank you very much for your review, based on it we have made detailed revisions to the article:
1., 2. and 4. We've made major revisions throughout the article.
3. The dataset was provided by the Department of Neurology in Zabrze, Medical University of Silesia, patients were anonymous and we were not present during their examination.
5. In the analyzed method, the number of patients is sufficient - the effectiveness of the mapping is checked separately for each model (unlike in the case of neural networks, where the accuracy is determined by the entire set).
Due to the fact that a large part of the article has changed, I am sending the entire article as an attachment.
Kind regards,
Joanna Chwał

Round 2
Reviewer 1 Report
I have formulated some recommendations for the basic version of the article:
1. The authors position their article as a research one, but they do not even bother to formulate the scientific novelty, object and subject of research. The design of the study would also be very appropriate. In addition, the topic is very hackneyed. A thoughtful, multi-criteria analysis of analogues with a list of their shortcomings is categorically necessary.
2. To be honest, it is not clear from the article what kind of result the authors got. I will proceed from the assumption that they were solving an image segmentation problem (what the authors say about 3D just makes the segmentation problem voluminous). So my hypothesis is built on the basis of formula (1) (this is one of the two formulas presented in the technical article). The most popular approach for the segmentation problem is to take some kind of neuroarchitecture (for example, Unet) with a pre-trained encoder and minimize the sum of $BCE$ (binary cross-entropy) and $DICE$ (Sørensen–Dice coefficient). The authors work with medical images. They are characterized by the presence of many defects. What problems do they create?
2a. Even if it seems to us that there are a lot of defects in a photograph, that it is very “dirty”, the area occupied by defects is still much smaller than the undamaged part of the image. To solve this problem, you can increase the weight of the positive class in $BCE$, and the optimal weight will be the ratio of the number of all clean pixels to the number of pixels belonging to defects.
2b. The second problem is that if we use Unet out of the box with a pre-trained encoder, like Albunet-18, we lose a lot of positional information. The first layer of Albunet-18 consists of a convolution with kernel 5 and stride equal to two. This allows the network to work quickly. We sacrificed network time for better defect localization: we removed max pooling after the first layer, reduced stride to 1, and reduced the convolution kernel to 3.
2c. If we work with small images, for example, compressing the picture to 256 x 256 or 512 x 512, then small defects will simply disappear due to interpolation. Therefore, you need to work with a large picture.
2d. During training, many pictures are placed on one card. Because of this, the estimation of the mean and variance in BatchNorm layers is inaccurate. In-place BatchNorm helps us solve this problem, which, firstly, saves memory, and secondly, it has a Synchronized BatchNorm version that synchronizes statistics between all cards.
3. However, judging by "3D Slicer (version 4.9.0), Blender (version 2.78c), Visual Studio 2017, Autodesk Meshmixer, Jupyter", we are not talking about intelligent image processing. The authors take a lot of images and build a 3D model "as God wills." I can't say otherwise, because the Methods section resembles a brochure, where the authors list the names without understanding the essence and rationality of building the model itself.
The author answered them in her own unique style. I do not consider her answers exhaustive. However, the article is not so bad. I stop at the "minor" version. Let the editor make the final decision.
Author Response
Dear Sir/Madam,
Thank you very much for your review. First of all, I would like to apologize for the imprecise answers to the questions asked. I have supplemented the previous notes with more precise answers, added in the attachment. Based on the comments and suggestions of the reviewers, we have once again introduced detailed corrections to the article.
Kind regards,
Joanna Chwał

Reviewer 2 Report
1. Despite my few suggestions, I do not see pseudo code of proposed method.
2. I also could not find the comparisons of proposed method.
3. Manuscript does not indicate how research gap is addressed.
4. Lack of mathematical analysis.
Author Response
Dear Sir/Madam,
Thank you very much for your review. I have added the answers to the questions in the attachment. Based on the comments and suggestions of the reviewers, we have once again introduced detailed corrections to the article.
Kind regards,
Joanna Chwał

Reviewer 3 Report
The revision has improved the quality of manuscript
Author Response
Dear Sir/Madam,
Thank you very much for your review. Based on the comments and suggestions of the other reviewers, we have once again introduced detailed corrections to the article.
Kind regards,
Joanna Chwał